# SIGNED GRAPH DIFFUSION NETWORK

## ABSTRACT

Given a signed social graph, how can we learn appropriate node representations to infer the signs of missing edges? Signed social graphs have received considerable attention to model trust relationships. Learning node representations is crucial to effectively analyze graph data, and various techniques such as network embedding and graph convolutional network (GCN) have been proposed for learning signed graphs. However, traditional network embedding methods are not end-to-end for a specific task such as link sign prediction, and GCN-based methods suffer from a performance degradation problem when their depth increases. In this paper, we propose SIGNED GRAPH DIFFUSION NETWORK (SGDNET), a novel graph neural network that achieves end-to-end node representation learning for link sign prediction in signed social graphs. We propose a random walk technique specially designed for signed graphs so that SGDNET effectively diffuses hidden node features. Through extensive experiments, we demonstrate that SGDNET outperforms state-of-the-art models in terms of link sign prediction accuracy.

## 1 INTRODUCTION

Given a signed social graph, how can we learn appropriate node representations to infer the signs of missing edges? Signed social graphs model trust relationships between people with positive (trust) and negative (distrust) edges. Many online social services such as Epinions (Guha et al., 2004) and Slashdot (Kunegis et al., 2009) that allow users to express their opinions are naturally represented as signed social graphs. Such graphs have attracted considerable attention for diverse applications including link sign prediction (Leskovec et al., 2010a; Kumar et al., 2016), node ranking (Jung et al., 2016; Li et al., 2019b), community analysis (Yang et al., 2007; Chu et al., 2016), graph generation (Derr et al., 2018a; Jung et al., 2020), and anomaly detection (Kumar et al., 2014). Node representation learning is a fundamental building block for analyzing graph data, and many researchers have put tremendous efforts into developing effective models for unsigned graphs. Graph convolutional networks (GCN) and their variants (Kipf & Welling, 2017; Velickovic et al., 2018) have spurred great attention in machine learning community, and recent works (Klicpera et al., 2019; Li et al., 2019a) have demonstrated stunning progress by handling the performance degradation caused by over-smoothing (Li et al., 2018; Oono & Suzuki, 2020) (i.e., node representations become indistinguishable as the number of propagation increases) or the vanishing gradient problem (Li et al., 2019a) in the first generation of GCN models. However, all of these models have a limited performance on node representation learning in signed graphs since they only consider unsigned edges under the homophily assumption (Kipf & Welling, 2017).

Many studies have been recently conducted to consider such signed edges, and they are categorized into network embedding and GCN-based models. Network embedding (Kim et al., 2018; Xu et al., 2019b) learns the representations of nodes by optimizing an unsupervised loss that primarily aims to locate two nodes' embeddings closely (or far) if they are positively (or negatively) connected. However, they are not trained jointly with a specific task in an end-to-end manner, i.e., latent features and the task are trained separately. Thus, their performance is limited unless each of them is tuned delicately. GCN-based models (Derr et al., 2018b; Li et al., 2020) have extended the graph convolutions to signed graphs using balance theory (Holland & Leinhardt, 1971) in order to properly propagate node features on signed edges. However, these models are directly extended from existing GCNs without consideration of the over-smoothing problem that degrades their performance. This problem hinders them from exploiting more information from multi-hop neighbors for learning node features in signed graphs.

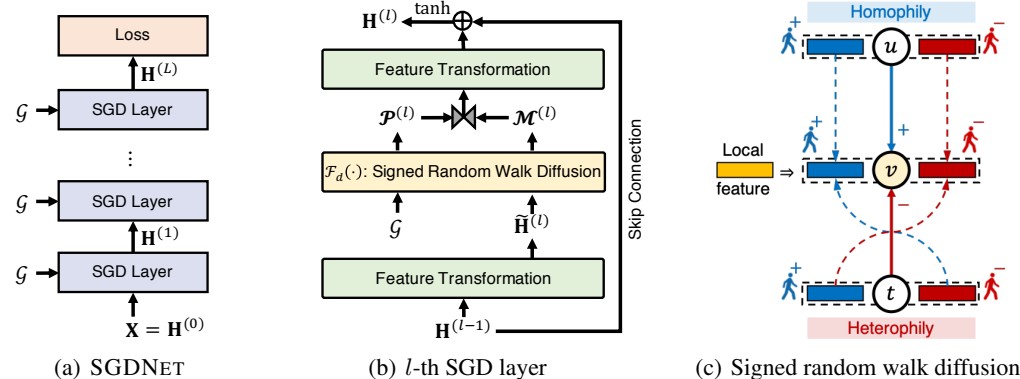

(a) SGDNET      (b) $l$-th SGD layer      (c) Signed random walk diffusion

Figure 1: Overall architecture of SGDNET. (a) Given a signed graph $\mathcal{G}$ and initial node features $\mathbf{X}$, SGDNET with multiple SGD layers produces the final embeddings $\mathbf{H}^{(L)}$, which is fed to a loss function under an end-to-end framework. (b) A single SGD layer learns node embeddings based on signed random walk diffusion. (c) Our diffusion module aggregates the features of node $v$ so that they are similar to those connected by $+$ edges (e.g., node $u$), and different from those connected by $-$ edges (e.g., node $t$). Also, it injects the local feature (i.e., the input feature of each module) of node $v$ at each aggregation to make the aggregated features distinguishable.

We propose SGDNET (SIGNED GRAPH DIFFUSION NETWORK), a novel graph neural network for node representation learning in signed graphs. Our main contributions are summarized as follows:

- **End-to-end learning.** We design SGDNET that performs end-to-end node representation learning. Given a signed graph, SGDNET produces node embeddings through multiple signed graph diffusion (SGD) layers (Figure 1(a)), which are fed into a loss function of a specific task such as link sign prediction.
- **Novel feature diffusion.** We propose a signed random walk diffusion method that propagates node embeddings on signed edges based on random walks considering signs, and injects local features (Figure 1(c)). This enables SGDNET to learn distinguishable node representations considering multi-hop neighbors while preserving local information.
- **Experiments.** Extensive experiments show that SGDNET effectively learns node representations of signed social graphs for link sign prediction, giving at least 3.9% higher accuracy than the state-of-the-art models in real datasets (Table 2).

## 2 RELATED WORK

### 2.1 GRAPH CONVOLUTIONAL NETWORKS ON UNSIGNED GRAPHS

Graph convolutional network (GCN) (Kipf & Welling, 2017) models the latent representation of a node by employing a convolutional operation on the features of its neighbors. Various GCN-based approaches (Kipf & Welling, 2017; Velickovic et al., 2018; Hamilton et al., 2017) have aroused considerable attention since they enable diverse graph supervised tasks (Kipf & Welling, 2017; Yao et al., 2019; Xu et al., 2019a) to be performed concisely under an end-to-end framework. However, the first generation of GCN models exhibit performance degradation due to the over-smoothing and vanishing gradient problems. Several works (Li et al., 2018; Oono & Suzuki, 2020) have theoretically revealed the over-smoothing problem. Also, Li et al. (Li et al., 2019a) have empirically shown that stacking more GCN layers leads to the vanishing gradient problem as in convolutional neural networks (He et al., 2016). Consequently, most GCN-based models (Kipf & Welling, 2017; Velickovic et al., 2018; Hamilton et al., 2017) are shallow; i.e., they do not use the feature information in faraway nodes when modeling node embeddings.

A recent research direction aims at resolving the limitation. Klicpera et al. (Klicpera et al., 2019) proposed APPNP exploiting Personalized PageRank (Jeh & Widom, 2003) to not only propagate hidden node embeddings far but also preserve local features, thereby preventing aggregated features from being over-smoothed. Li et al. (Li et al., 2019a) suggested ResGCN adding skip connections between GCN layers, as in ResNet (He et al., 2016). However, all of these models do not provide how to use signed edges since they are based on the homophily assumption (Kipf & Welling, 2017), i.e.,

users having connections are likely to be similar, which is not valid for negative edges. As opposed to the homophily, negative edges have the semantics of heterophily (Rogers, 2010), i.e., users having connections are dissimilar. Although these methods can still be applied to signed graphs by ignoring the edge signs, their trained features have limited capacity.

## 2.2 Network Embedding and Graph Convolutional Networks on Signed Graphs

Traditional methods on network embedding extract latent node features specialized for signed graphs in an unsupervised manner. Kim et al. (Kim et al., 2018) proposed SIDE which optimizes a likelihood over direct and indirect signed connections on truncated random walks sampled from a signed graph. Xu et al. (Xu et al., 2019b) developed SLF considering positive, negative, and non-linked relationships between nodes to learn non-negative node embeddings. However, such approaches are not end-to-end, i.e., they are not directly optimized for solving a supervised task such as link prediction.

There are recent progresses on end-to-end learning on signed networks under the GCN framework. Derr et al. (Derr et al., 2018b) proposed SGCN which extends the GCN mechanism to signed graphs considering balanced and unbalanced relationships supported by structural balance theory (Holland & Leinhardt, 1971). Yu et al. (Li et al., 2020) developed SNEA using attention techniques to reveal the importance of these relationships. However, such state-of-the-art models do not consider the over-smoothing problem since they are directly extended from GCN.

## 3 Proposed Method

We propose SGDNET (SIGNED GRAPH DIFFUSION NETWORK), a novel end-to-end model for node representation learning in signed graphs. Our SGDNET aims to properly aggregate node features on signed edges, and to effectively use the features of multi-hop neighbors so that generated features are not over-smoothed. Our main ideas are to diffuse node features along random walks considering the signs of edges, and to inject local node features at each aggregation.

Figure 1 depicts the overall architecture of SGDNET. Given a signed graph $\mathcal{G}$ and initial node features $\mathbf{X} \in \mathbb{R}^{n \times d_0}$ as shown in Figure 1(a), SGDNET extracts the final node embeddings $\mathbf{H}^{(L)} \in \mathbb{R}^{n \times d_L}$ through multiple SGD layers where $n$ is the number of nodes, $L$ is the number of SGD layers, and $d_l$ is the embedding dimension of the $l$-th layer. Then, $\mathbf{H}^{(L)}$ is fed into a loss function of a specific task so that they are jointly trained in an end-to-end framework. Given $\mathbf{H}^{(l-1)}$, the $l$-th SGD layer aims to learn $\mathbf{H}^{(l)}$ based on feature transformations and signed random walk diffusion $\mathcal{F}_d(\cdot)$ as shown in Figure 1(b). The layer also uses the skip connection to prevent the vanishing gradient problem when the depth of SGDNET increases.

Figure 1(c) illustrates the intuition behind the signed random walk diffusion. Each node has two features corresponding to positive and negative surfers, respectively. The surfer flips its sign when moving along negative edges, while the sign is kept along positive edges. For example, the positive (or negative) surfer becomes positive at node $v$ if it moves from a positively connected node $u$ (or a negatively connected node $t$). As a result, the aggregated features at node $v$ become similar to those connected by positive edges (e.g., node $u$), and different from those connected by negative edges (e.g., node $t$). In other words, it satisfies homophily and heterophily at the same time while unsigned GCNs cannot handle the heterophily of negative edges. Furthermore, we inject the local feature (i.e., the input feature of the module) of node $v$ at each aggregation so that the resulting features remain distinguishable during the diffusion.

## 3.1 Signed Graph Diffusion Layer

Given a signed graph $\mathcal{G}$ and the node embeddings $\mathbf{H}^{(l-1)}$ from the previous layer, the $l$-th SGD layer learns new embeddings $\mathbf{H}^{(l)}$ as shown in Figure 1(b). It first transforms $\mathbf{H}^{(l-1)}$ into hidden features $\tilde{\mathbf{H}}^{(l)}$ as $\tilde{\mathbf{H}}^{(l)} = \mathbf{H}^{(l-1)}\mathbf{W}_t^{(l)}$ with a learnable parameter $\mathbf{W}_t^{(l)} \in \mathbb{R}^{d_{l-1} \times d_l}$. Then, it applies the signed random walk diffusion which is represented as the function $\mathcal{F}_d(\mathcal{G}, \tilde{\mathbf{H}}^{(l)})$ that returns $\boldsymbol{\mathcal{P}}^{(l)} \in \mathbb{R}^{n \times d_l}$ and $\boldsymbol{\mathcal{M}}^{(l)} \in \mathbb{R}^{n \times d_l}$ as the positive and the negative embeddings, respectively (details in Section 3.2). The embeddings are concatenated and transformed as follows:

$$\mathbf{H}^{(l)} = \phi\left(\left[\boldsymbol{\mathcal{P}}^{(l)} \| \boldsymbol{\mathcal{M}}^{(l)}\right] \mathbf{W}_n^{(l)} + \mathbf{H}^{(l-1)}\right) \tag{1}$$

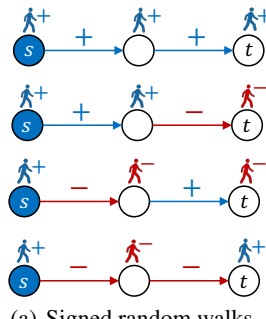

(a) Signed random walks

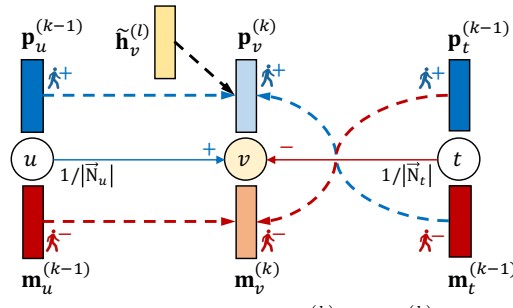

(b) Feature diffusion for $\mathbf{p}_v^{(k)}$ and $\mathbf{m}_v^{(k)}$

Figure 2: Feature diffusion by signed random walks in SGDNET. (a) Signed random walks properly consider edge signs. (b) The positive and the negative feature vectors $\mathbf{p}_v^{(k)}$ and $\mathbf{m}_v^{(k)}$ are updated from the previous feature vectors and the local feature vector $\tilde{\mathbf{h}}_v^{(l)}$ as described in Equation (2).

where $\phi(\cdot)$ is a non-linear activator such as `tanh`, $\|$ denotes horizontal concatenation of two matrices, and $\mathbf{W}_n^{(l)} \in \mathbb{R}^{2d_l \times d_l}$ is a trainable weight matrix that learns a relationship between $\mathcal{P}^{(l)}$ and $\mathcal{M}^{(l)}$. We use the skip connection (He et al., 2016; Li et al., 2019a) with $\mathbf{H}^{(l-1)}$ in Equation (1) to avoid the vanishing gradient issue which frequently occurs when multiple layers are stacked.

## 3.2 SIGNED RANDOM WALK DIFFUSION

We design the signed random walk diffusion operator $\mathcal{F}_d(\cdot)$ used in the $l$-th SGD layer. Given the signed graph $\mathcal{G}$ and the hidden node embeddings $\tilde{\mathbf{H}}^{(l)}$, the diffusion operator $\mathcal{F}_d(\cdot)$ diffuses the node features based on random walks considering edge signs so that it properly aggregates node features on signed edges and prevents the aggregated features from being over-smoothed.

Signed random walks are performed by a signed random surfer (Jung et al., 2016) who has the $+$ or $-$ sign when moving around the graph. Figure 2(a) shows signed random walks on four cases according to edge signs: 1) a friend's friend, 2) a friend's enemy, 3) an enemy's friend, and 4) an enemy's enemy. The surfer starts from node $s$ with the $+$ sign. If it encounters a negative edge, the surfer flips its sign from $+$ to $-$, or vice versa. Otherwise, the sign is kept. The surfer determines whether a target node $t$ is a friend of node $s$ or not according to its sign.

$\mathcal{F}_d(\cdot)$ exploits the signed random walk for diffusing node features on signed edges. Each node is represented by two feature vectors which represent the positive and negative signs, respectively. Let $k$ denote the number of diffusion steps or random walk steps. Then, $\mathbf{p}_v^{(k)} \in \mathbb{R}^{d_l \times 1}$ and $\mathbf{m}_v^{(k)} \in \mathbb{R}^{d_l \times 1}$ are aggregated at node $v$, respectively, where $\mathbf{p}_v^{(k)}$ (or $\mathbf{m}_v^{(k)}$) is the feature vector visited by the positive (or negative) surfer at step $k$. These are recursively obtained by the following equations:

$$\mathbf{p}_v^{(k)} = (1-c)\left( \sum_{u \in \overleftarrow{\mathrm{N}}_v^+} \frac{1}{|\overrightarrow{\mathrm{N}}_u|}\mathbf{p}_u^{(k-1)} + \sum_{t \in \overleftarrow{\mathrm{N}}_v^-} \frac{1}{|\overrightarrow{\mathrm{N}}_t|}\mathbf{m}_t^{(k-1)} \right) + c\tilde{\mathbf{h}}_v^{(l)}$$

$$\mathbf{m}_v^{(k)} = (1-c)\left( \sum_{t \in \overleftarrow{\mathrm{N}}_v^-} \frac{1}{|\overrightarrow{\mathrm{N}}_t|}\mathbf{p}_t^{(k-1)} + \sum_{u \in \overleftarrow{\mathrm{N}}_v^+} \frac{1}{|\overrightarrow{\mathrm{N}}_u|}\mathbf{m}_u^{(k-1)} \right)$$

(2)

where $\overleftarrow{\mathrm{N}}_v^s$ is the set of incoming neighbors to node $v$ connected with edges of sign $s$, $\overrightarrow{\mathrm{N}}_u$ is the set of outgoing neighbors from node $u$ regardless of edge signs, $\tilde{\mathbf{h}}_v^{(l)}$ is the local feature of node $v$ (i.e., the $v$-th row vector of $\tilde{\mathbf{H}}^{(l)}$), and $0 < c < 1$ is a local feature injection ratio. That is, the features are computed by the signed random walk feature diffusion with weight $1 - c$ and the local feature injection with weight $c$ with the following details.

**Signed Random Walk Feature Diffusion.** Figure 2(b) illustrates how $\mathbf{p}_v^{(k)}$ and $\mathbf{m}_v^{(k)}$ are diffused by the signed random walks according to Equation (2). Suppose the positive surfer visits node $v$ at step $k$. For this to happen, the positive surfer of an incoming neighbor $u$ at step $k-1$ should choose the edge $(u \to v, +)$ by a probability $1/|\overrightarrow{\mathrm{N}}_u|$. This transition to node $v$ along the positive edge allows to keep the surfer's positive sign. At the same time, the negative surfer of an incoming neighbor $t$ at step $k-1$ should move along the edge $(t \to v, -)$ by a probability $1/|\overrightarrow{\mathrm{N}}_t|$. In this case, the surfer

flips its sign from $-$ to $+$. Considering these signed random walks, $\mathbf{p}_v^{(k)}$ is obtained by the weighted aggregation of $\mathbf{p}_u^{(k-1)}$ and $\mathbf{m}_t^{(k-1)}$. Similarly, $\mathbf{m}_v^{(k)}$ is aggregated as shown in Figure 2(b).

**Local Feature Injection.** Although the feature diffusion above properly considers edge signs, the generated features could be over-smoothed after many steps if we depend solely on the diffusion. In other words, it considers only the graph information explored by the signed random surfer, while the local information in the hidden feature $\tilde{\mathbf{h}}_v^{(l)}$ is disregarded during the diffusion. Hence, as shown in Figure 2(b), we explicitly inject the local feature $\tilde{\mathbf{h}}_v^{(l)}$ to $\mathbf{p}_v^{(k)}$ with weight $c$ at each aggregation in Equation (2) so that the diffused features are not over-smoothed. The reason why local features are only injected to $+$ embeddings is that we consider a node should trust ($+$) its own information (i.e., its local feature).

### 3.3 Convergence Guarantee of Signed Random Walk Diffusion

Suppose that $\mathbf{P}^{(k)} = [\mathbf{p}_1^{(k)\top}; \cdots ; \mathbf{p}_n^{(k)\top}]$ and $\mathbf{M}^{(k)} = [\mathbf{m}_1^{(k)\top}; \cdots ; \mathbf{m}_n^{(k)\top}]$ represent the positive and negative embeddings of all nodes, respectively, where ; denotes vertical concatenation. Let $\mathbf{A}_s$ be the adjacency matrix for sign $s$ such that $\mathbf{A}_{suv}$ is 1 for signed edge $(u \to v, s)$, and 0 otherwise. Then, Equation (2) is vectorized as follows:

$$\left.\begin{aligned}\mathbf{P}^{(k)} &= (1-c)(\tilde{\mathbf{A}}_+^\top \mathbf{P}^{(k-1)} + \tilde{\mathbf{A}}_-^\top \mathbf{M}^{(k-1)}) + c\tilde{\mathbf{H}}^{(l)}\\ \mathbf{M}^{(k)} &= (1-c)(\tilde{\mathbf{A}}_-^\top \mathbf{P}^{(k-1)} + \tilde{\mathbf{A}}_+^\top \mathbf{M}^{(k-1)})\end{aligned}\right\} \Rightarrow \mathbf{T}^{(k)} = (1-c)\tilde{\mathbf{B}}\mathbf{T}^{(k-1)} + c\mathbf{Q} \quad (3)$$

where $\tilde{\mathbf{A}}_s = \mathbf{D}^{-1}\mathbf{A}_s$ is the normalized matrix for sign $s$, and $\mathbf{D}$ is a diagonal out-degree matrix (i.e., $\mathbf{D}_{ii} = |\vec{\mathbf{N}}_i|$). The left equation of Equation (3) is compactly represented as the right equation where

$$\mathbf{T}^{(k)} = \begin{bmatrix}\mathbf{P}^{(k)}\\ \mathbf{M}^{(k)}\end{bmatrix} \qquad \tilde{\mathbf{B}} = \begin{bmatrix}\tilde{\mathbf{A}}_+^\top & \tilde{\mathbf{A}}_-^\top\\ \tilde{\mathbf{A}}_-^\top & \tilde{\mathbf{A}}_+^\top\end{bmatrix} \qquad \mathbf{Q} = \begin{bmatrix}\tilde{\mathbf{H}}^{(l)}\\ \mathbf{0}\end{bmatrix}.$$

Then, $\mathbf{T}^{(k)}$ is guaranteed to converge as shown in the following theorem.

**Theorem 1** *The diffused features in $\mathbf{T}^{(k)}$ converge to equilibrium for $c \in (0,1)$ as follows:*

$$\mathbf{T}^* = \lim_{k\to\infty}\mathbf{T}^{(k)} = \lim_{k\to\infty}\left(\sum_{i=0}^{k-1}(1-c)^i\tilde{\mathbf{B}}^i\right)\tilde{\mathbf{Q}} = (\mathbf{I} - (1-c)\tilde{\mathbf{B}})^{-1}\tilde{\mathbf{Q}} \qquad (\tilde{\mathbf{Q}} := c\mathbf{Q}) \quad (4)$$

*If we iterate Equation (3) $K$ times for $1 \le k \le K$, the exact solution $\mathbf{T}^*$ is approximated as*

$$\mathbf{T}^* \approx \mathbf{T}^{(K)} = \tilde{\mathbf{Q}} + (1-c)\tilde{\mathbf{B}}\tilde{\mathbf{Q}} + \cdots + (1-c)^{K-1}\tilde{\mathbf{B}}^{K-1}\tilde{\mathbf{Q}} + (1-c)^K\tilde{\mathbf{B}}^K\mathbf{T}^{(0)} \quad (5)$$

*where $\|\mathbf{T}^* - \mathbf{T}^{(K)}\|_1 \le (1-c)^K\|\mathbf{T}^* - \mathbf{T}^{(0)}\|_1$, and $\mathbf{T}^{(0)} = \begin{bmatrix}\mathbf{P}^{(0)}\\ \mathbf{M}^{(0)}\end{bmatrix}$ is the initial value of Equation (3).* □

**Proof 1** *A proof sketch is to show the spectral radius of $\tilde{\mathbf{B}}$ is less than or equal to 1, which guarantees the convergence of the geometric series with $(1-c)\tilde{\mathbf{B}}$. See the details in Appendix A.1.* □

According to Theorem 1, $\tilde{\mathbf{B}}^K\tilde{\mathbf{Q}}$ is the node features diffused by $K$-step signed random walks with $\tilde{\mathbf{Q}}$ where $\tilde{\mathbf{B}}^K$ is interpreted as the transition matrix of $K$-step signed random walks. Thus, the approximation is the sum of the diffused features from 1 to $K$ steps with a decaying factor $1 - c$, i.e., the effect of distant nodes gradually decreases while that of neighboring nodes is high. This is the reason why SGDNET prevents diffused features from being over-smoothed. Also, the approximation error $\|\mathbf{T}^* - \mathbf{T}^{(K)}\|_1$ exponentially deceases as $K$ increases due to the term $(1-c)^K$. Another point is that the iteration of Equation (3) converges to the same solution no matter what $\mathbf{P}^{(0)}$ and $\mathbf{M}^{(0)}$ are given. In this work, we initialize $\mathbf{P}^{(0)}$ with $\tilde{\mathbf{H}}^{(l)}$, and randomly initialize $\mathbf{M}^{(0)}$ in $[-1, 1]$.

The signed random walk diffusion operator $\mathcal{F}_d(\cdot)$ iterates Equation (3) $K$ times for $1 \le k \le K$ where $K$ is the number of diffusion steps, and it returns $\mathcal{P}^{(l)} \leftarrow \mathbf{P}^{(K)}$ and $\mathcal{M}^{(l)} \leftarrow \mathbf{M}^{(K)}$ as the outputs of the diffusion module at the $l$-th SGD layer. The detailed pseudocode of SGDNET is described in Appendix A.3, and its time complexity is analyzed in Appendix A.2.

Table 1: Dataset statistics of signed graphs. |V| and |E| are the number of nodes and edges, respectively. Given sign $s \in \{+, -\}$, $|E^s|$ and $\rho(s)$ are the number and percentage of edges with sign $s$, respectively.

| Dataset | $\|V\|$ | $\|E\|$ | $\|E^+\|$ | $\|E^-\|$ | $\rho(+)$ | $\rho(-)$ |
|---|---|---|---|---|---|---|
| Bitcoin-Alpha[1] | 3,783 | 24,186 | 22,650 | 1,536 | 93.65% | 6.35% |
| Bitcoin-OTC[1] | 5,881 | 35,592 | 32,029 | 3,563 | 89.99% | 10.01% |
| Slashdot[2] | 79,120 | 515,397 | 392,326 | 123,071 | 76.12% | 23.88% |
| Epinions[3] | 131,828 | 841,372 | 717,667 | 123,705 | 85.30% | 14.70% |

[1] `https://snap.stanford.edu/data/soc-sign-bitcoin-otc.html`
[2] `http://konect.uni-koblenz.de/networks/slashdot-zoo`
[3] `http://www.trustlet.org/wiki/Extended_Epinions_dataset`

### 3.4 LOSS FUNCTION FOR LINK SIGN PREDICTION

The link sign prediction is to predict the missing sign of a given edge. As shown in Figure 1(a), SGDNET produces the final node embeddings $\mathbf{H}^{(L)}$. The embeddings are fed into a loss function $\mathcal{L}(\mathcal{G}, \mathbf{H}^{(L)}; \mathbf{\Theta}) = \mathcal{L}_{sign}(\mathcal{G}, \mathbf{H}^{(L)}) + \lambda \mathcal{L}_{reg}(\mathbf{\Theta})$ where $\mathbf{\Theta}$ is the set of model parameters, $\mathcal{L}_{sign}(\cdot)$ is the binary cross entropy loss, and $\mathcal{L}_{reg}(\cdot)$ is the $L_2$ regularization loss with weight decay $\lambda$. For a signed edge $(u \rightarrow v, s)$, the edge feature is $\mathbf{z}_{uv} \in \mathbb{R}^{1 \times 2d_L} = \mathbf{h}_u^{(L)} || \mathbf{h}_v^{(L)}$ where $\mathbf{h}_u^{(L)}$ is the $u$-th row vector of $\mathbf{H}^{(L)}$. Let E be the set of signed edges. Then, $\mathcal{L}_{sign}(\cdot)$ is represented as follows:

$$\mathcal{L}_{sign}(\mathcal{G}, \mathbf{X}) = - \sum_{(u \rightarrow v, s) \in E} \sum_{t \in \{+, -\}} \mathbb{I}(t = s) \log \left( \texttt{softmax}_t \left( \mathbf{z}_{uv} \mathbf{W} \right) \right)$$

where $\mathbf{W} \in \mathbb{R}^{2d_L \times 2}$ is a learnable weight matrix, $\texttt{softmax}_t(\cdot)$ is the probability for sign $t$ after softmax operation, and $\mathbb{I}(\cdot)$ returns 1 if a given predicate is true, and 0 otherwise.

## 4 EXPERIMENTS

We evaluate the effectiveness of SGDNET through the link sign prediction task.

**Datasets.** We perform experiments on four standard signed graphs summarized in Table 1: Bitcoin-Alpha (Kumar et al., 2016), Bitcoin-OTC (Kumar et al., 2016), Slashdot (Kunegis et al., 2009), and Epinions (Guha et al., 2004). We provide the detailed description of each dataset in Appendix A.4. We also report additional experiments on Wikipedia dataset (Leskovec et al., 2010b) in Appendix A.5.

**Competitors.** We compare our proposed SGDNET with the following competitors:

- **APPNP** (Klicpera et al., 2019): an unsigned GCN model based on Personalized PageRank.
- **ResGCN** (Li et al., 2019a): another unsigned GCN model exploiting skip connections to deeply stack multiple layers.
- **SIDE** (Kim et al., 2018): a network embedding model optimizing the likelihood over signed edges using random walk sequences to encode structural information into node embeddings.
- **SLF** (Xu et al., 2019b): another network embedding model considering positive, negative, and non-linked relationships to learn non-negative node embeddings.
- **SGCN** (Derr et al., 2018b): a state-of-the-art signed GCN model considering balanced and unbalanced paths motivated from balance theory to propagate embeddings.
- **SNEA** (Li et al., 2020): another signed GCN model extending SGCN by learning attentions on the balanced and unbalanced paths.

We use the absolute adjacency matrix for APPNP and ResGCN since they handle only unsigned edges. All methods are implemented by PyTorch and Numpy in Python. We use a machine with Intel E5-2630 v4 2.2GHz CPU and Geforce GTX 1080 Ti for the experiments.

**Evaluation Metrics.** We randomly split the edges of a signed graph into training and test sets by the 8:2 ratio. As shown in Table 1, the sign ratio is highly skewed to the positive sign, i.e., the sampled datasets are naturally imbalanced. Considering the class imbalance, we measure the area under the curve (AUC) to evaluate predictive performance. We also report F1-macro measuring the average of the ratios of correct predictions for each sign since negative edges need to be treated as important as positive edges (i.e., it gives equal importance to each class). A higher value of AUC or F1-macro indicates better performance. We repeat each experiment 10 times with different random seeds and report the average and standard deviation of test values.

Table 2: SGDNET gives the best link sign prediction performance in terms of AUC. The best model is in bold, and the second best model is underlined. The % increase measures the best accuracy against the second best accuracy.

| AUC | Bitcoin-Alpha | Bitcoin-OTC | Slashdot | Epinions |
|---|---|---|---|---|
| **APPNP** (Klicpera et al., 2019) | 0.854±0.010 | 0.867±0.009 | 0.837±0.003 | 0.870±0.002 |
| **ResGCN** (Li et al., 2019a) | 0.853±0.017 | 0.876±0.010 | 0.744±0.004 | 0.871±0.002 |
| **SIDE** (Kim et al., 2018) | 0.801±0.020 | 0.839±0.013 | 0.814±0.003 | 0.880±0.003 |
| **SLF** (Xu et al., 2019b) | 0.779±0.023 | 0.797±0.014 | 0.833±0.006 | 0.876±0.005 |
| **SGCN** (Derr et al., 2018b) | 0.824±0.018 | 0.857±0.008 | 0.827±0.004 | 0.895±0.002 |
| **SNEA** (Li et al., 2020) | 0.855±0.006 | 0.858±0.008 | 0.754±0.005 | 0.771±0.004 |
| **SGDNet (proposed)** | **0.911±0.007** | **0.921±0.005** | **0.886±0.001** | **0.932±0.001** |
| % increase | 6.4% | 4.9% | 5.9% | 3.9% |

Table 3: SGDNET gives the best link sign prediction performance in terms of F1-macro.

| F1-macro | Bitcoin-Alpha | Bitcoin-OTC | Slashdot | Epinions |
|---|---|---|---|---|
| **APPNP** (Klicpera et al., 2019) | 0.682±0.005 | 0.762±0.009 | 0.748±0.003 | 0.773±0.004 |
| **ResGCN** (Li et al., 2019a) | 0.658±0.006 | 0.735±0.015 | 0.609±0.004 | 0.784±0.003 |
| **SIDE** (Kim et al., 2018) | 0.663±0.008 | 0.709±0.008 | 0.685±0.009 | 0.785±0.006 |
| **SLF** (Xu et al., 2019b) | 0.615±0.027 | 0.641±0.025 | 0.733±0.008 | 0.810±0.008 |
| **SGCN** (Derr et al., 2018b) | 0.690±0.014 | 0.776±0.008 | 0.752±0.013 | 0.844±0.002 |
| **SNEA** (Li et al., 2020) | 0.670±0.005 | 0.742±0.011 | 0.690±0.005 | 0.805±0.005 |
| **SGDNet (proposed)** | **0.757±0.012** | **0.799±0.007** | **0.778±0.002** | **0.854±0.002** |
| % increase | 7.4% | 1.6% | 3.5% | 1.2% |

**Hyperparameter Settings.** We set the dimension of final node embeddings to 32 for all methods so that their embeddings have the same learning capacity (see its effect in Appendix A.6). We perform 5-fold cross-validation for each method to find the best hyperparameters and measure the test accuracy with the selected ones. In the cross-validation for SGDNET, the number $L$ of SGD layers is sought between 1 and 6, and the restart probability $c$ is selected from 0.05 to 0.95 by step size 0.1. We set the number $K$ of diffusion steps to 10 and the feature dimension $d_l$ of each layer to 32. We follow the range of each hyperparameter recommended in its corresponding paper for the cross-validation of other models. Our model is trained by the Adam optimizer (Kingma & Ba, 2015), where the learning rate is 0.01, the weight decay $\lambda$ is 0.001, and the number of epochs is 100. We summarize the hyperparameters used by SGDNET for each dataset in Appendix A.7.

## 4.1 LINK SIGN PREDICTION

We evaluate the performance of each method on link sign prediction. Tables 2 and 3 summarize the experimental results in terms of AUC and F1-macro, respectively. Note that our SGDNET shows the best performance in terms of AUC and F1-macro scores. SGDNET presents $3.9 \sim 6.4\%$ and $1.2 \sim 7.4\%$ improvements over the second best models in terms of AUC and F1-macro, respectively. We have the following observations.

- The unsigned GCN models APPNP and ResGCN show worse performance than SGDNET, which shows the importance of using sign information.
- The performance of network embedding techniques such as SIDE and SLF is worse than that of other GCN-based models; this shows the importance of jointly learning feature extraction and link sign prediction for the performance.
- The performance of SGCN and SNEA which use limited features from nodes within $2 \sim 3$ hops is worse than that of SGDNET which exploits up to $K$-hop neighbors' features where $K$ is set to 10 in these experiments. It indicates that carefully exploiting features from distant nodes as well as neighboring ones is crucial for the performance.

## 4.2 EFFECT OF DIFFUSION STEPS

We investigate the effect of the feature diffusion in SGDNET for learning signed graphs. We use one SGD layer, and set the restart probability $c$ to 0.15 to evaluate the pure effect of the diffusion

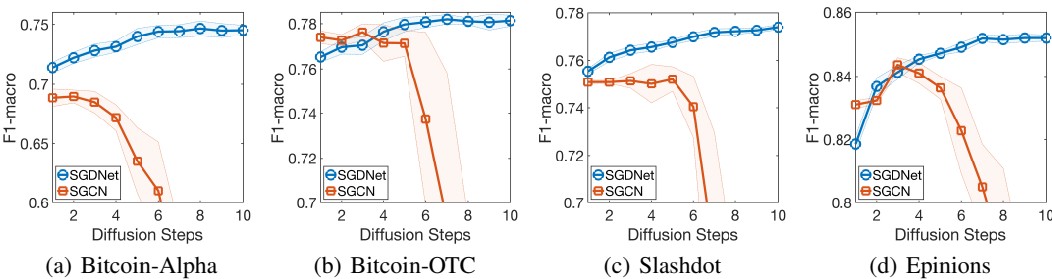

(a) Bitcoin-Alpha      (b) Bitcoin-OTC      (c) Slashdot      (d) Epinions

Figure 3: Effect of SGDNET's feature diffusion compared to state-of-the-art SGCN. The performance of SGDNET is boosted while that of SGCN degrades as the number $K$ of diffusion steps increases.

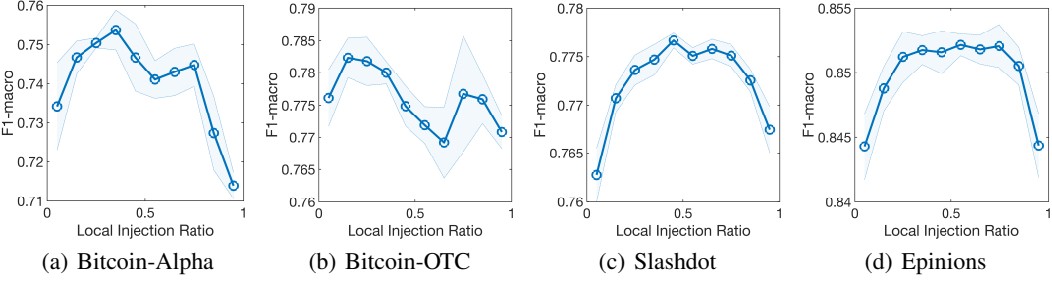

(a) Bitcoin-Alpha      (b) Bitcoin-OTC      (c) Slashdot      (d) Epinions

Figure 4: Effect of local injection ratio $c$ of SGDNET. A relatively small value ($0.15 \sim 0.35$) of $c$ is the best for the Bitcoin-Alpha and Bitcoin-OTC (small) datasets while $c$ around $0.5$ shows better accuracy for the Slashdot and Epinions (large) datasets.

module; we vary the number $K$ of diffusion steps from $1$ to $10$ and evaluate the performance of SGDNET in terms of F1-macro for each diffusion step. Also, we compare SGDNET to SGCN, a state-of-the-art-model for learning signed graphs. The number of diffusion steps of SGCN is determined by its number of layers. Figure 3 shows that the performance of SGDNET gradually improves as $K$ increases while that of SGCN dramatically decreases over all datasets. This indicates that SGCN suffers from the performance degradation problem when its network becomes deep, i.e., it is difficult to use more information beyond 3 hops in SGCN. On the other hand, SGDNET utilizes features of farther nodes, and generates more expressive and stable features than SGCN does. Note that the performance of SGDNET converges in general after a sufficient number of diffusion steps, which is highly associated with Theorem 1.

## 4.3 EFFECT OF LOCAL INJECTION RATIO

We examine the effect of the local injection ratio $c$ in the diffusion module of SGDNET. We use one SGD layer, and set the number $K$ of diffusion steps to $10$; we vary $c$ from $0.05$ to $0.95$ by $0.1$, and measure the performance of the link sign prediction task in terms of F1-macro. Figure 4 shows the effect of $c$ to the predictive performance of SGDNET. For small datasets such as Bitcoin-Alpha and Bitcoin-OTC, $c$ between $0.15$ and $0.35$ provides better performance. On the other hand, $c$ around $0.5$ shows higher accuracy for large datasets such as Slashdot and Epinions. For all datasets, a too low or too high value of $c$ (e.g., $0.05$ or $0.95$) results in a poor performance.

## 5 CONCLUSION

In this paper, we propose SIGNED GRAPH DIFFUSION NETWORK (SGDNET), a novel graph neural network that performs end-to-end node representation learning for link sign prediction in signed graphs. We propose a signed random walk diffusion method to properly diffuse node features on signed edges, and suggest a local feature injection method to make diffused features distinguishable. Our diffusion method empowers SGDNET to effectively train node embeddings considering multi-hop neighbors while preserving local information. Our extensive experiments show that SGDNET provides the best accuracy outperforming the state-of-the-art models in link sign prediction. Future research directions include extending our method for multi-view networks.

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

# A APPENDIX

## A.1 CONVERGENCE ANALYSIS

**Theorem 1 (Convergence of Signed Random Walk Diffusion)** *The diffused features in $\mathbf{T}^{(k)}$ converge to equilibrium for $c \in (0, 1)$ as follows:*

$$\mathbf{T}^* = \lim_{k \to \infty} \mathbf{T}^{(k)} = \lim_{k \to \infty} \left( \sum_{i=0}^{k-1} (1-c)^i \tilde{\mathbf{B}}^i \right) \tilde{\mathbf{Q}} = (\mathbf{I} - (1-c)\tilde{\mathbf{B}})^{-1} \tilde{\mathbf{Q}} \qquad (\tilde{\mathbf{Q}} \coloneqq c\mathbf{Q})$$

*If we iterate Equation (3) $K$ times for $1 \le k \le K$, the exact solution $\mathbf{T}^*$ is approximated as*

$$\mathbf{T}^* \approx \mathbf{T}^{(K)} = \tilde{\mathbf{Q}} + (1-c)\tilde{\mathbf{B}}\tilde{\mathbf{Q}} + \cdots + (1-c)^{K-1}\tilde{\mathbf{B}}^{K-1}\tilde{\mathbf{Q}} + (1-c)^K \tilde{\mathbf{B}}^K \mathbf{T}^{(0)}$$

*where $\|\mathbf{T}^* - \mathbf{T}^{(K)}\|_1 \le (1-c)^K \|\mathbf{T}^* - \mathbf{T}^{(0)}\|_1$, and $\mathbf{T}^{(0)} = \begin{bmatrix} \mathbf{P}^{(0)} \\ \mathbf{M}^{(0)} \end{bmatrix}$ is the initial value of Equation (3).* □

**Proof 1** *The iteration of Equation (3) is written as follows:*

$$\begin{aligned}
\mathbf{T}^{(k)} &= (1-c)\tilde{\mathbf{B}}\mathbf{T}^{(k-1)} + c\mathbf{Q} \\
&= \left( (1-c)\tilde{\mathbf{B}} \right)^2 \mathbf{T}^{(k-2)} + \left( (1-c)\tilde{\mathbf{B}} + \mathbf{I} \right) \tilde{\mathbf{Q}} \\
&= \cdots \\
&= \left( (1-c)\tilde{\mathbf{B}} \right)^k \mathbf{T}^{(0)} + \left( \sum_{i=0}^{k-1} \left( (1-c)^i \tilde{\mathbf{B}}^i \right) \right) \tilde{\mathbf{Q}}.
\end{aligned} \qquad (6)$$

*Note that the spectral radius $\rho(\tilde{\mathbf{B}})$ is less than or equal to 1 by Theorem 2; thus, for $0 < c < 1$, the spectral radius of $(1-c)\tilde{\mathbf{B}}$ is less than 1, i.e., $\rho((1-c)\tilde{\mathbf{B}}) = (1-c)\rho(\tilde{\mathbf{B}}) \le (1-c) < 1$. Hence, if $k \to \infty$, the power of $(1-c)\tilde{\mathbf{B}}$ converges to $\mathbf{0}$, i.e., $\lim_{k \to \infty}(1-c)^k \tilde{\mathbf{B}}^k = \mathbf{0}$. Also, the second term in Equation (6) becomes the infinite geometric series of $(1-c)\tilde{\mathbf{B}}$ which converges as the following equation:*

$$\mathbf{T}^* = \lim_{k \to \infty} \mathbf{T}^{(k)} = \mathbf{0} + \lim_{k \to \infty} \left( \sum_{i=0}^{k-1} \left( (1-c)^i \tilde{\mathbf{B}}^i \right) \right) \tilde{\mathbf{Q}} = (\mathbf{I} - (1-c)\tilde{\mathbf{B}})^{-1} \tilde{\mathbf{Q}}$$

*where the convergence always holds if $\rho((1-c)\tilde{\mathbf{B}}) < 1$. The converged solution $\mathbf{T}^*$ satisfies $\mathbf{T}^* = (1-c)\tilde{\mathbf{B}}\mathbf{T}^* + c\mathbf{Q}$. Also, $\mathbf{T}^*$ is approximated as Equation (5). Then, the approximation error $\|\mathbf{T}^* - \mathbf{T}^{(K)}\|_1$ is bounded as follows:*

$$\begin{aligned}
\|\mathbf{T}^* - \mathbf{T}^{(K)}\|_1 &= \|(1-c)\tilde{\mathbf{B}}\mathbf{T}^* - (1-c)\tilde{\mathbf{B}}\mathbf{T}^{(K-1)}\|_1 \le (1-c)\|\tilde{\mathbf{B}}\|_1 \|\mathbf{T}^* - \mathbf{T}^{(K-1)}\|_1 \\
&\le (1-c)\|\mathbf{T}^* - \mathbf{T}^{(K-1)}\|_1 \le \cdots \\
&\le (1-c)^K \|\mathbf{T}^* - \mathbf{T}^{(0)}\|_1
\end{aligned} \qquad (7)$$

*where $\|\cdot\|_1$ is $L_1$-norm of a matrix. Note that the bound $\|\tilde{\mathbf{B}}\|_1 \le 1$ of Theorem 2 is used in the above derivation.* □

**Theorem 2 (Bound of Spectral Radius of $\tilde{\mathbf{B}}$)** *The spectral radius of $\tilde{\mathbf{B}}$ in Equation (3) is less than or equal to 1, i.e., $\rho(\tilde{\mathbf{B}}) \le \|\tilde{\mathbf{B}}\|_1 \le 1$.* □

**Proof 2** *According to spectral radius theorem (Trefethen & Bau III, 1997), $\rho(\tilde{\mathbf{B}}) \le \|\tilde{\mathbf{B}}\|_1$ where $\|\cdot\|_1$ denotes $L_1$-norm of a given matrix, indicating the maximum absolute column sum of the matrix. Note that the entries of $\tilde{\mathbf{B}}$ are non-negative probabilities; thus, the absolute column sums of $\tilde{\mathbf{B}}$ are equal to its column sums which are obtained as follows:*

$$\mathbf{1}_{2n}^\top \tilde{\mathbf{B}} = \begin{bmatrix} \mathbf{1}_n^\top \tilde{\mathbf{A}}_+^\top + \mathbf{1}_n^\top \tilde{\mathbf{A}}_-^\top & \mathbf{1}_n^\top \tilde{\mathbf{A}}_-^\top + \mathbf{1}_n^\top \tilde{\mathbf{A}}_+^\top \end{bmatrix} = \begin{bmatrix} \mathbf{1}_n^\top \tilde{\mathbf{A}}^\top & \mathbf{1}_n^\top \tilde{\mathbf{A}}^\top \end{bmatrix} = \begin{bmatrix} \mathbf{b}^\top & \mathbf{b}^\top \end{bmatrix} \qquad (8)$$

where $\tilde{\mathbf{A}}^\top = \tilde{\mathbf{A}}_+^\top + \tilde{\mathbf{A}}_-^\top$, and $\mathbf{1}_n$ is an n-dimensional one vector. Note that $\tilde{\mathbf{A}}_s^\top = \mathbf{A}_s^\top \mathbf{D}^{-1}$ for sign s where $\mathbf{D}$ is a diagonal out-degree matrix (i.e., $\mathbf{D}_{uu} = |\vec{\mathbf{N}}_u|$). Then, $\mathbf{1}_n^\top \tilde{\mathbf{A}}^\top$ is represented as

$$\mathbf{1}_n^\top \tilde{\mathbf{A}}^\top = \mathbf{1}_n^\top (\mathbf{A}_+^\top + \mathbf{A}_-^\top) \mathbf{D}^{-1} = \mathbf{1}_n^\top |\mathbf{A}|^\top \mathbf{D}^{-1} = (|\mathbf{A}|\mathbf{1}_n)^\top \mathbf{D}^{-1} = \mathbf{b}^\top$$

where $|\mathbf{A}| = \mathbf{A}_+ + \mathbf{A}_-$ is the absolute adjacency matrix. The u-th entry of $|\mathbf{A}|\mathbf{1}_n$ indicates the out-degree of node u, denoted by $|\vec{\mathbf{N}}_u|$. Note that $\mathbf{D}_{uu}^{-1}$ is $1/|\vec{\mathbf{N}}_u|$ if u is a non-deadend. Otherwise, $\mathbf{D}_{uu}^{-1} = 0$ (i.e., a deadend node has no outgoing edges). Hence, the u-th entry of $\mathbf{b}^\top$ is 1 if node u is not a deadend, or 0 otherwise; its maximum value is less than or equal to 1. Therefore, $\rho(\tilde{\mathbf{B}}) \le \|\tilde{\mathbf{B}}\|_1 \le 1$. □

## A.2 Time Complexity Analysis

**Theorem 3 (Time Complexity of SGDNET)** *The time complexity of the l-th SGD layer is $O(Kmd_l + nd_{l-1}d_l)$ where K is the number of diffusion steps, $d_l$ is the feature dimension of the l-th layer, and m and n are the number of edges and nodes, respectively. Assuming all of $d_l$ are set to d, SGDNET with L SGD layers takes $O(LKmd + Lnd^2)$ time.* □

**Proof 3** *The feature transform operations require $O(nd_{l-1}d_l)$ time due to their dense matrix multiplication. Each iteration of the signed random walk diffusion in Equation (3) takes $O(md_l)$ time due to the sparse matrix multiplication $\tilde{\mathbf{B}}\mathbf{T}^{(k-1)}$ where the number of non-zeros of $\tilde{\mathbf{B}}$ is $O(m)$. Thus, $O(Kmd_l)$ is required for K iterations. Overall, the total time complexity of the l-th SGD layer is $O(Kmd_l + nd_{l-1}d_l)$.* □

## A.3 Pseudocode of SGDNET

Algorithm 1 describes SGDNET's overall procedure which is depicted in Figure 1. Given signed adjacency matrix $\mathbf{A}$ and related hyper-parameters (e.g., numbers L and K of SGD layers and diffusion steps, respectively), SGDNET produces the final hidden node features $\mathbf{H}^{(L)}$ which are fed to a loss function as described in Section 3.4. It first computes the normalized matrices $\tilde{\mathbf{A}}_+$ and $\tilde{\mathbf{A}}_-$ (line 1). Then, it performs the forward function of SGDNET (lines $3 \sim 12$). The forward function repeats the signed random walk diffusion K times (lines $6 \sim 9$), and then performs the non-linear feature transformation skip-connected with $\mathbf{H}^{(l-1)}$ (line 11).

---

**Algorithm 1:** Pseudocode of SGDNET

---

**Input:** signed adjacency matrix $\mathbf{A}$, initial node feature matrix $\mathbf{X}$, number K of diffusion steps, number L of SGD layers, and local feature injection ratio c
**Output:** hidden node feature matrix $\mathbf{H}^{(L)}$

1: compute normalized matrices for each sign, i.e., $\tilde{\mathbf{A}}_+ = \mathbf{D}^{-1}\mathbf{A}_+$ and $\tilde{\mathbf{A}}_- = \mathbf{D}^{-1}\mathbf{A}_-$
2: initialize $\mathbf{H}^{(0)}$ with $\mathbf{X}$
3: **for** $l \leftarrow 1$ to L **do**                    ▷ *start the forward function of* SGDNET
4:     perform the feature transformation as $\tilde{\mathbf{H}}^{(l)} \leftarrow \mathbf{H}^{(l-1)}\mathbf{W}_t^{(l)}$
5:     initialize $\mathbf{P}^{(0)}$ with $\tilde{\mathbf{H}}^{(l)}$ and randomly initialized $\mathbf{M}^{(0)}$ in $[-1, 1]$
6:     **for** $k \leftarrow 1$ to K **do**           ▷ *perform the signed random walk diffusion in Equation* (2)
7:         $\mathbf{P}^{(k)} \leftarrow (1-c)(\tilde{\mathbf{A}}_+^\top \mathbf{P}^{(k-1)} + \tilde{\mathbf{A}}_-^\top \mathbf{M}^{(k-1)}) + c\tilde{\mathbf{H}}^{(l)}$
8:         $\mathbf{M}^{(k)} \leftarrow (1-c)(\tilde{\mathbf{A}}_-^\top \mathbf{P}^{(k-1)} + \tilde{\mathbf{A}}_+^\top \mathbf{M}^{(k-1)})$
9:     **end for**
10:    set $\boldsymbol{\mathcal{P}}^{(l)} \leftarrow \mathbf{P}^{(K)}$ and $\boldsymbol{\mathcal{M}}^{(l)} \leftarrow \mathbf{M}^{(K)}$
11:    compute l-th hidden node features $\mathbf{H}^{(l)} \leftarrow \tanh([\boldsymbol{\mathcal{P}}^{(l)}\|\boldsymbol{\mathcal{M}}^{(l)}]\mathbf{W}_n^{(l)} + \mathbf{H}^{(l-1)})$
12: **end for**
13: **return** $\mathbf{H}^{(L)}$

---

### A.4 DETAILED DESCRIPTION OF DATASETS

The Bitcoin-Alpha and Bitcoin-OTC datasets (Kumar et al., 2016) are extracted from directed online trust networks served by Bitcoin Alpha and Bitcoin OTC, respectively. The Slashdot dataset (Kunegis et al., 2009) is collected from Slashdot, a technology news site which allows a user to create positive or negative links to others. The Epinions dataset (Guha et al., 2004) is a directed signed graph scraped from Epinions, a product review site in which users mark their trust or distrust to others.

The publicly available signed graphs do not contain initial node features even though they have been utilized as standard datasets in signed graph analysis. Due to this reason, many previous works (Derr et al., 2018b; Li et al., 2020) on GCN for signed graphs have exploited singular vector decomposition (SVD) to extract initial node features. Thus, we follow this setup, i.e., $\mathbf{X} = \mathbf{U}\mathbf{\Sigma}_d$ is the initial feature matrix for all GCN-based models where $\mathbf{A} \simeq \mathbf{U}\mathbf{\Sigma}_{d_i}\mathbf{V}^\top$ is obtained by a truncated SVD method, called Randomized SVD (Halko et al., 2011), with target rank $d_i = 128$. Note that the method is very efficient (i.e., its time complexity is $O(nd_i^2)$ where $n$ is the number of nodes) and performed only once as a preprocessing in advance; thus, it does not affect the computational performance of training and inference.

### A.5 ADDITIONAL EXPERIMENTS ON WIKIPEDIA DATASET

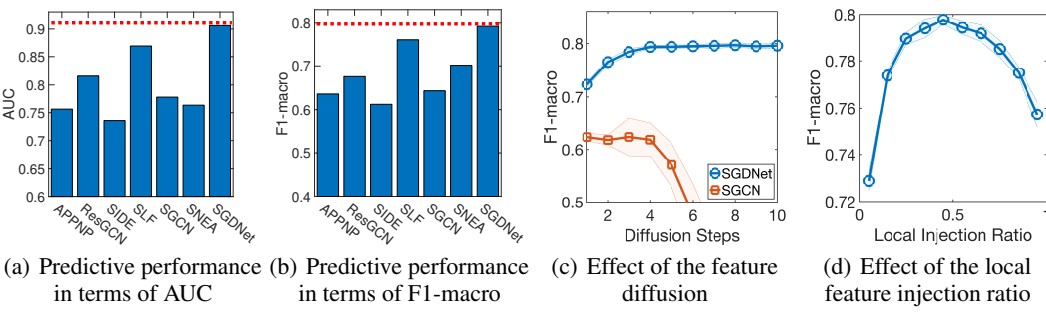

(a) Predictive performance in terms of AUC     (b) Predictive performance in terms of F1-macro     (c) Effect of the feature diffusion     (d) Effect of the local feature injection ratio

Figure 5: Experimental results on Wikipedia dataset.

We perform additional experiments on Wikipedia dataset (Leskovec et al., 2010b) which has been also frequently used in signed graph analysis. The Wikipedia dataset is a signed graph representing the administrator election procedure in Wikipedia where a user can vote for ($+$) or against ($-$) a candidate. The numbers of nodes and edges are $7,118$ and $103,675$, respectively. Figure 5 shows the experimental results on the dataset. As seen in Figures 5(a) and 5(b), SGDNET outperforms other methods in terms of AUC and F1-macro, respectively. Figure 5(c) indicates our diffusion mechanism still works on the Wikipedia dataset. Figure 5(d) shows the effect of the local feature injection ratio $c$, indicating properly selected $c$ such as $0.5$ is helpful for the performance.

### A.6 EFFECT OF EMBEDDING DIMENSION

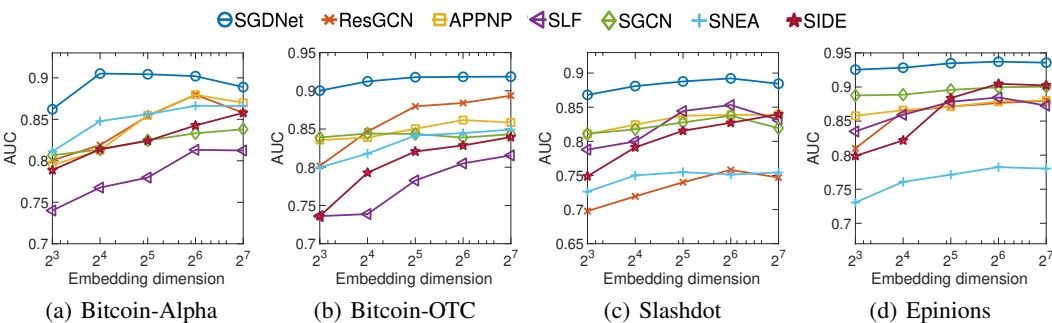

(a) Bitcoin-Alpha     (b) Bitcoin-OTC     (c) Slashdot     (d) Epinions

Figure 6: Effect of the embedding dimension of each model.

We investigate the effect of the node embedding dimension of each model in the datasets listed in Table 1. For this experiment, we vary the dimension of hidden and final node embeddings from 8 to 128, and observe the trend of AUC in the link sign prediction task. As shown in Figure 6, SGDNET outperforms its competitors over all the tested dimensions, and it is relatively less sensitive to the embedding dimension than other models in all datasets except Bitcoin-Alpha.

## A.7 HYPERPARAMETER CONFIGURATION

Table 4: We summarize the configurations of SGDNET's hyperparameters, which are used in the experiments of this paper.

| Hyperparameter | Bitcoin-Alpha | Bitcoin-OTC | Slashdot | Epinions | Wikipedia |
|---|---|---|---|---|---|
| Number $L$ of SGD layers | 1 | 2 | 2 | 2 | 2 |
| Local injection ratio $c$ | 0.35 | 0.25 | 0.55 | 0.55 | 0.5 |
| Number $K$ of diffusion steps | | | 10 | | |
| Input feature dimension $d_i$ | | | 128 | | |
| Hidden embedding dimension $d_l$ | | | 32 | | |
| Optimizer | | Adam (learning rate: 0.01, weight decay $\lambda$: 0.001) | | | |
| Number of epochs | | | 100 | | |

