# OpenReview forum: "Signed Graph Diffusion Network"
_ICLR.cc/2021/Conference — Reject_

### Official Review · AnonReviewer2 · 2020-10-26
**The paper propose SGDNet, Signed Graph Diffusion Network, to perform end-to-end node presentation and signed link prediction. It also considers the over-smoothing issue and use local feature injection to prevent over-smoothing.**

**Rating:** 4
**Confidence:** 3

**Review:**

The strength:

- Empirical results show improvements of the proposed method over the baseline methods, including methods for unsigned graphs, signed embedding emthods, and signed GCN methods not addressing over-smoothing.

The weakness:

- The proposed method is a straightforward integration of existing methods on signed graphs and on handling over-smoothing in GCNs. There is very little new idea in the proposed method.

- The convergence result (Theorem 1) is straightforward based on linear algebra, and it is only for the diffusion part in each GCN layer. There is no overall theoretical results on the SGDNet architecture.

Overall, the method proposed is a simple combination of past methods on handling signed graphs and handling over-smoothing in graphs. It is just that the two has not been combined together, and so the only contribution I see is the combination of these two methods and verify it in the signed edge prediction task. The theoretical result on convergence is only on the diffusion convergence at each layer, and it is a straightforward application of the linear algebra. It is unclear to me why we need a diffusion convergence at each layer and then also need GCN with multiple layers. What is the connection between the diffusion steps K and the GCN layers L?

In summary, the paper shows improvement when combining previous methods on signed networks and on handling over-smoothing. It may fit into a second-tier conference to record the result, but I feel that it does not meet the high bar of ICLR.

---

> ### Author Response · Authors · 2020-11-21
> **Response to AnonReviewer2**
>
> We deeply appreciate your dedicated review of our paper. As the reviewer mentioned, our model is inspired by two existing methods: SRWR [1] and APPNP [2].  As an unsupervised approach, SRWR computes trustworthiness scores between nodes in signed graphs. The paper of APPNP addressed the over-smoothing issue by utilizing Personalized PageRank in plain networks. As the review pointed out, our model is built upon the combination of the two methods in a high-level language. However, we do not think that our contributions are limited to such abstract summary. It is not trivial to reveal why the signed random walks, which were used in SRWR with a different purpose, are effective for node feature aggregation, and why the combination alleviates the over-smoothing problem when it comes to representation learning in signed graphs. Thus, our genuine contributions come from our answers (e.g., Figure 1(c) and Equation (5) in Theorem 1) for the aforementioned questions within the combination.
>
> Apart from this, we want to point out that in a high-level language, SRWR is a combination of random walks and structural balance theory, and APPNP is summarized by a combination of Personalized PageRank and GCN from the perspective of the reviewer. Each of them was an existing method at that time, and each integration is straightforward. Nevertheless, the papers of SRWR and APPNP have been published at a top-tier conference after showing that each combination is effective for its own purpose and task.  As many papers including the above papers did, we think this is one way that human knowledge incrementally evolves over time, and we believe that our approaches and outcomes are beneficial to signed graph analysis communities.
>
> Although the reviewer said our convergence theorem is straightforward, it is not trivial to show the necessary condition that enables our diffusion model to converge. The condition is that the spectral radius of B <= 1, and we theoretically proved it in Theorem 2 of Appendix A.1. The convergence theorem explains why our model prevents diffused features from being over-smoothed, and it is likely to make diffused features stable for the performance of a target task (note that the standard deviation of our model is mostly small in Tables 2 and 3). Also, the theorem implies several numerical advantages, i.e., we do not need to infinitely repeat the diffusion since the approximation error monotonically and exponentially decreases.
>
> The main reason why we use multiple SGD layers is that we aim to increase learning capacity by modeling latent non-linear relationships captured by tanh between diffused features of adjacent SGD layers with more parameters. The connection between the diffusion steps K and the GCN layers L is that our model performs K * L-hop feature propagations with O(L) trainable weight matrices where each SGD layer addresses the over-smoothing problem within K diffusion steps. Note that if we train a weight matrix for each hop propagation, the model becomes too complicated and heavy; thus, the performance will degrade due to overfitting.
>
> [1] Jung, J., Jin, W., Sael, L., & Kang, U. (2016, December). Personalized ranking in signed networks using signed random walk with restart. In 2016 IEEE 16th International Conference on Data Mining (ICDM) (pp. 973-978). IEEE.
>
> [2] Klicpera, J., Bojchevski, A., & Günnemann, S. (2018, September). Predict then Propagate: Graph Neural Networks meet Personalized PageRank. In International Conference on Learning Representations.

---

### Official Review · AnonReviewer1 · 2020-10-27
**Official Blind Review #1**

**Rating:** 6
**Confidence:** 4

**Review:**

Main Idea:

In this paper, the author studied the problem of node embedding in signed networks. The authors proposed SGDNet which combines the idea of diffusion/random work in signed networks and Residual connection in GCN. The network is trained directly with classification loss on edge sign prediction. The authors carried out extensive experiments on several real-world networks with comparison to several state-of-the-art methods. The proposed method showed superior performance in the sign prediction task.

Strength:
The paper is technically sound and well written. The authors provide analysis on the convergence of the diffusion kernel to show that the injection of input signals prevents the degradation with depth increment.
The authors carried out thorough experiments on four real-world networks with comparison to several strong baselines.
The authors carried out parameter sensitivity analysis on two of the important parameters: diffusion depth and signal injection strength.

Weakness
The authors trained the model and evaluated performance only on existing edges with signs. However, in practice, it is usually useful to predict the existence of edges or not in the network. It is interesting to see how the method generalizes to this situation which more resembles the graph reconstruction task.
The comparison of depth in SGCN with K is kind of unfair. A more direct comparison would be the number of layers L which controls the depth of the network. Since there are no trainable parameters involved in the diffusion step.
The authors mentioned that the embedding dimension is fixed to 32 for all methods. First, as an important hyperparameter, it is interesting to see how performance of different methods varies with it. Second, the embedding dimension of the proposed method is effective 64 as positive and negative are separated and combined.

Questions
In equation (2), what is the motivation to inject h only to positive but not the negative as well.
How is m^{(0)}_t and p^{(0)}_t initialized. Are they initialized to all zero?

---

> ### Author Response · Authors · 2020-11-21
> **Response to AnonReviewer1**
>
> Thank you so much for your devoted and huge effort in reviewing our paper.
>
> **Generalization to graph reconstruction task**
>
> Our model is also able to easily generalize to the graph reconstruction task by just replacing the final loss function of SGDNet with RMSE loss for the existence of signed edges. This is one of the strengths of end-to-end learning that our model pursues. As the reviewer mentioned, we acknowledge that it is proper to evaluate methods for node representation learning in signed graphs, and we will consider this as our future work. The main reason why we chose the sign prediction task is that it has been considered and evaluated as a standard task in related papers of state-of-the-art methods such as SIDE, SLF, SGCN, and SNEA.
>
> **The comparison of depth in SGCN**
>
> First of all, we want to clarify that Figure 3 aims to show the performance change depending on hops that features are propagated.  From this aspect, the hop count is determined by the depth in SGCN while it is decided by the diffusion steps in our model. In other words, each propagation and learning are tightly coupled in SGCN while they are decoupled in SGDNet. This freedom makes our model easy to use more features from distant nodes under our mechanism for preventing over-smoothing. Thus, we wanted to emphasize that using more trainable parameters is not always good, and such freedom from the decoupling is beneficial to multi-hop feature aggregations.
>
> **Effect of embedding dimension**
>
> The final embedding indicates the embedding of an “edge” which is obtained by concatenating positive and negative embeddings (i.e., $\mathbf{z}_{uv}$ in Section 3.4). Thus, we make the final embeddings generated by all tested models have the same dimension 32, i.e., the dimension of the final embeddings of our model is 32, not 64. We will clarify it in Section 4 of the revised paper.
> Also, you can check the effect of the embedding dimension in Appendix A.6 of the submitted paper. As seen in Figure 6, our model outperforms its competitors over all the tested dimensions, and it is relatively less sensitive to the embedding dimension than other models for all datasets except Bitcoin-Alpha.
>
> **Motivation on local injection**
>
> As described in Section 3.2, the reason why local features are only injected to + embeddings is that we presume a node should trust (+) its own information (i.e., its local feature).
>
> **How is m^{(0)}_t and p^{(0)}_t initialized?**
>
> As described in Section 3.3, we initialize $\mathbf{P}^{(0)}$ with $\mathbf{H}^{(l)}$ , and randomly initialize $\mathbf{M}^{(0)}$ in [−1, 1] in this work. However, how to initialize them is not that important since the iteration of Equation (3) converges to the same solution no matter what $\mathbf{P}^{(0)}$ and $\mathbf{M}^{(0)}$ are given (see Theorem 1).

---

### Official Review · AnonReviewer3 · 2020-10-29
**Graph Diffusion Network on signed networks**

**Rating:** 4
**Confidence:** 4

**Review:**

The paper proposes to leverage signed random walk as hidden representation propagations to construct a signed graph diffusion network model.

Pros:
1. The idea of the proposed model is simple but very effective according to the experiment evaluations.
2. It's very interesting that the proposed model achieves better performance as the number of layers increases, even up to K=10.
3. The writing of the paper is in a good shape and everything is clear to follow.
4. The authors also prove the convergence of the proposed diffusion layer.

Cons:
1. My major concern is the novelty of the paper. The key part of the model (i.e., signed diffusion layer) totally borrows from the existing work. Even the figure used in this paper is same as the reference paper (e.g., Figure 2 (a) in this paper vs. Figure 2 (b) in reference paper).
      Jung, Jinhong, et al. "Personalized ranking in signed networks using signed random walk with restart." 2016 IEEE 16th
      International Conference on Data Mining (ICDM). IEEE, 2016.
2. In addition, authors need to compare with the above paper as well as it's very effective in link prediction even though it's not an embedding-based method.

---

> ### Author Response · Authors · 2020-11-21
> **Response to AnonReviewer3**
>
> We are genuinely grateful for your review of our paper. First of all, we need to clarify that Figure 2(a) is used just for depicting the concept of signed random walks proposed in the reference paper, i.e., we do not propose the signed random walks (also, we cited the paper in Section 3.2). In the reference paper, the signed random walks are proposed for propagating probabilities on each node, not d-dimensional node features, to measure node-to-node similarity scores.
>
> On the other hand, our novel contribution is to have shown that node features are also effectively propagated through the signed random walks as seen in Figure 2(b) of the submitted paper. Furthermore, we have shown how the signed random walks are able to satisfy both homophily and heterophily between nodes in Figure 1(c). The reference paper cannot answer how to propagate multidimensional node features, and provide why the signed random walks work well on node feature aggregation for end-to-end learning in signed graphs.
>
> Hence, we claim that our work is distinguishable from the reference paper, and each paper has its distinct contributions. They have different purposes, i.e., the reference paper aims to design node similarity, while our goal is to develop a new way for node feature aggregation in signed graphs.  Our work has shown that the signed random walks also work well on node representation learning in signed graphs. To clearly deliver the concept of the signed random walks to readers, we borrowed the figure mentioned by the reviewer from the reference paper under the agreement of the paper’s authors.
>
> As mentioned above, the target problem of our paper is different from that of the reference paper (SRWR) although both approaches share the concept of the signed random walks. Our problem aims to model latent node representations in an end-to-end framework while the problem in the reference paper is to compute trustworthiness scores between nodes in an unsupervised manner. Due to this reason, we considered other STOA models such as SGCN and SNEA for node representation learning as our competitors, rather than similarity measures such as SRWR in signed graphs. Of course, as mentioned by the reviewer, we can add an experiment to compare our model with SRWR since both utilized the sign prediction task for evaluation. However, for node representation learning, it is hard to answer which of SGDNet and SRWR models better representation because SRWR does not produce latent node features.

---

### Official Review · AnonReviewer4 · 2020-11-01
**Signed network embedding**

**Rating:** 7
**Confidence:** 3

**Review:**

This paper presents an interesting new graph neural network technique customized for the specific setting of signed graphs. This is a paper I have reviewed before.

There has been an increasing interest in the last several years on the problem of clustering/link prediction/representation learning of signed graphs, where edge weights are allowed to take either positive or negative values. The main contribution is the end to end pipeline targeted at link sign prediction and the feature diffusion step. The signed random walk diffusion is neat, and appears to be new I believe, in this context. The numerical results are somewhat convincing, with improvements in the single digit percentages; on some data sets fairly small, but consistent improvement across the board. There is some theoretical component, which is rather light, but this is fine.

Might have been good to compare against non GCN based methods, for example
Chiang, Kai-Yang, et al. "Prediction and clustering in signed networks: a local to global perspective." The Journal of Machine Learning Research 15.1 (2014): 1177-1213.
Perhaps also worth placing this work in the context of some of the growing literature of signed clustering/link prediction that typically rely on spectral methods/matrix completion. However, since the authors already compare against non-GCN methods and it is fine to leave this for future work.

It would have been good to see a comparison on synthetic data as well. There exist signed stochastic block models which have been looked at, which could be augmented with features, or use no feature information at all. Another compelling story could be made around better clustering performance, by leveraging the embedding obtained via the authors’ proposed approach. There is a growing literature on clustering signed graphs, and it would be interesting to see to what extend the proposed method embedding is applicable in that setting and how cluster recovery performance looks like.

Regarding the features - they play a central role in the paper, but towards the end the reader finds out that the input signed graphs do not have initial node features.. It would be have been good to identify a data set where the nodes come with some covariate information. Or, as stated above, a synthetic model could be put forth, with a set (or subset) of features which correlate with cluster membership/edge sign.

In SBM model, it would be interesting to see how does the link prediction accuracy vary with the number of clusters, and also the graph sparsity. Most spectral methods out there for this task face difficulties when handling very sparse graphs (ie, typically below the threshold connectivity limit of (log n)/n, both theoretically and empirically, and require various regularisation techniques). It would be good for the authors to comment on this aspect (very sparse graphs), as it could be a strong edge over spectral methods which tend to underperform in the very space regime.

Might also be interesting to comment on how seed information could be used, if available? See for example some of the literature on polarization in signed graphs.

Overall the paper appears to be technically sound, clearly written and very well structured, and I think overall a good addition to the signed network embedding literature.

---

> ### Author Response · Authors · 2020-11-21
> **Response to AnonReviewer4**
>
> We really appreciate your constructive and valuable feedback on our paper. As suggested by the reviewer, one of our future steps is to investigate our approach for the clustering problem in signed networks as the importance of the topic continues to grow, and it has received considerable attention. During the research on this topic, it will be beneficial to examine the tendency of performance according to various graph properties and features controlled by the aforementioned SBM.

---

### Decision · Program_Chairs · 2021-01-07
**Final Decision**

**Decision:**

Reject

**Comment:**

The paper addresses an interesting problem of clustering/link prediction/representation learning of signed graphs, where edge weights are allowed to take either positive or negative values. The paper proposed an end to end pipeline targeted at link sign prediction and the feature diffusion step. The reviewers think the proposed method is a straightforward integration of existing methods, and the convergence result is straightforward. The paper can be improved by including more novel ideas or analysis.